# Multitarget-Tracking Method Based on the Fusion of Millimeter-Wave Radar and LiDAR Sensor Information for Autonomous Vehicles

**DOI:** 10.3390/s23156920

**Published:** 2023-08-03

**Authors:** Junren Shi, Yingjie Tang, Jun Gao, Changhao Piao, Zhongquan Wang

**Affiliations:** 1School of Automation, Chongqing University of Posts and Telecommunications, Chongqing 400065, China; laoauhako@gmail.com (Y.T.); piaoch@cqupt.edu.cn (C.P.); akaxi611@gmail.com (Z.W.); 2School of Computer Science and Technology, Chongqing University of Posts and Telecommunications, Chongqing 400065, China; gaoj315@gmail.com

**Keywords:** millimeter-wave radar, lidar, autonomous vehicles, multitarget tracking, data fusion

## Abstract

Multitarget tracking based on multisensor fusion perception is one of the key technologies to realize the intelligent driving of automobiles and has become a research hotspot in the field of intelligent driving. However, most current autonomous-vehicle target-tracking methods based on the fusion of millimeter-wave radar and lidar information struggle to guarantee accuracy and reliability in the measured data, and cannot effectively solve the multitarget-tracking problem in complex scenes. In view of this, based on the distributed multisensor multitarget tracking (DMMT) system, this paper proposes a multitarget-tracking method for autonomous vehicles that comprehensively considers key technologies such as target tracking, sensor registration, track association, and data fusion based on millimeter-wave radar and lidar. First, a single-sensor multitarget-tracking method suitable for millimeter-wave radar and lidar is proposed to form the respective target tracks; second, the Kalman filter temporal registration method and the residual bias estimation spatial registration method are used to realize the temporal and spatial registration of millimeter-wave radar and lidar data; third, use the sequential m-best method based on the new target density to find the track the correlation of different sensors; and finally, the IF heterogeneous sensor fusion algorithm is used to optimally combine the track information provided by millimeter-wave radar and lidar, and finally form a stable and high-precision global track. In order to verify the proposed method, a multitarget-tracking simulation verification in a high-speed scene is carried out. The results show that the multitarget-tracking method proposed in this paper can realize the track tracking of multiple target vehicles in high-speed driving scenarios. Compared with a single-radar tracker, the position, velocity, size, and direction estimation errors of the track fusion tracker are reduced by 85.5%, 64.6%, 75.3%, and 9.5% respectively, and the average value of GOSPA indicators is reduced by 19.8%; more accurate target state information can be obtained than a single-radar tracker.

## 1. Introduction

Smart cars have great potential in reducing traffic accidents, alleviating traffic congestion, and improving road and vehicle utilization. Their main technologies can be divided into three parts: environmental perception [1], path planning [2], and decision-making control [3]. As a basic work in the environment perception of intelligent vehicles, multitarget tracking [4] is of great significance for ensuring the safety of autonomous vehicles and improving the ability of intelligent vehicles to understand the environment.

Millimeter-wave radars and LiDAR have become mainstream sensors on board autonomous vehicles for target tracking [5]. However, millimeter-wave radars cannot obtain the geometric information and category information of the target, and are easily affected by clutter, noise, and multipath, and false targets in the detection also limit its multitarget-tracking capability [6]. Compared with millimeter-wave radar, lidar can obtain three-dimensional information of the driving environment and can give very accurate spatial position information. It has incomparable advantages in obstacle detection and target ranging, but the long-distance point cloud is sparse. As a result, its multitarget-tracking capability is limited [7]. In order to solve the multitarget-tracking problem in complex scenes, it is imperative to extend single-sensor multitarget tracking to multisensor multitarget tracking.

Multisensor target-tracking technology is also called multisensor information fusion technology, which can be divided into centralized, distributed, and hybrid according to the architecture [8]. Among them, the centralized information fusion system is to send the original measurement information obtained by each sensor to the central processor for time–space registration, data association, tracking, and other processing. This fusion method has a small information loss and high data fusion accuracy, which can reach fusion in the optimal sense [9,10,11,12]. However, centralized fusion requires high data quality and large communication bandwidth, which will increase the burden on the fusion center and result in poor real-time data processing. Each sensor in the distributed information fusion system has its own data-processing center, which can independently process the obtained measurement data. Each local sensor sends the processed and compressed data to the fusion center, where they are combined and reasoned, and information fusion is finally realized. Compared with centralized fusion, distributed fusion has lower requirements on channel capacity, strong fault tolerance, and easy expansion [13,14,15,16]. The hybrid information fusion model requires each sensor to send the original measurement and the processed local target track to the fusion center at the same time, taking the advantages of both centralized and distributed fusion into account, but the structure of the hybrid fusion method is more complex than the previous two fusion methods, increasing the cost of communication and calculation [17,18,19,20]. Because the distributed fusion structure is simple and easy to implement, this paper chooses the distributed fusion structure as the basic framework for autonomous-vehicle multitarget tracking.

As an important research direction in multisource information fusion, target tracking is an important way to realize sensor network detection. The change in the number of targets and the uncertainty of measurement information bring great challenges to multitarget tracking. In traditional multitarget-tracking algorithms based on data association technology, such as the Joint Probability Data Association (JPDA) algorithm [21], Multiple Hypotheses Tracking (MHT) algorithm [22], etc., as the number of targets and the number of clutters increase, the number of calculations increases exponentially. The Probability Hypothesis Density (PHD) filter [23] based on a Random Finite Set (RFS) can achieve target without complex data association by passing the first-order moment of the multitarget posterior probability density estimation of the number and target state. In the process of distributed multisensor multitarget tracking, it is necessary to transform the data from multiple sensors into the same space–time reference frame. Since different sensors have different transmission rates and sampling periods, and there are sensor system deviations and measurement errors, direct conversion will reduce the accuracy of data fusion. Therefore, sensor spatiotemporal registration is required when processing multisensor data [24]. In the distributed multisensor multitarget tracking system, each source has an independent information-processing system, which can independently track the surrounding environment targets and generate corresponding target tracks. Tracks from different systems may represent the same target due to overlapping detection areas between sensors. Therefore, track correlation between sensors is required to find the track corresponding to the same target [25]. Distributed multisensor multitarget tracking is also called distributed multisensor data fusion. In this system, each local sensor first forms its own target track based on the single-sensor multitarget-tracking algorithm, and then each sensor sends the target track to the fusion center to complete the space–time registration and track association, and then the fusion center based on the estimated fusion criterion tracks from the same target are estimated and fused to form a stable, high-precision global track [26].

At present, most autonomous-vehicle target-tracking methods based on the fusion of millimeter-wave radar and lidar information rarely consider key technologies such as target tracking, sensor registration, track correlation, and data fusion, and it is difficult to guarantee the accuracy and reliability of the data with these methods. They cannot effectively solve the multitarget-tracking problem in complex scenes.

In conclusion, this research makes the following contributions:

Taking the distributed multisensor multitarget tracking (DMMT) system as the basic framework and based on millimeter-wave radar and lidar, a multitarget-tracking method for autonomous vehicles is proposed that comprehensively considers key technologies such as target tracking, sensor registration, track correlation, data fusion, etc., so as to effectively improve the target-tracking accuracy and stability in complex scenes. The main work is as follows: (i) taking millimeter-wave radar and lidar as objects, respectively, a single-sensor multitarget-tracking method suitable for millimeter-wave radar and lidar is proposed to form their respective target tracks; (ii) using the Kalman filter temporal registration method and the residual bias estimation spatial registration method to realize the space–time registration of millimeter-wave radar and lidar data; (iii) using the sequential m-best method based on the new target density to find the track correlation of different sensors; (iv) using the IF heterogeneous sensor fusion algorithm to optimally combine the track information provided by millimeter-wave radar and lidar, and finally form a stable and high-precision global track.

## 2. Program Framework

The research program in this paper is mainly for autonomous vehicles under high-speed driving conditions. Its main characteristics include: high driving speed (higher than 80 km/h), many traffic participants, and complex driving scenarios. When an autonomous vehicle is driving on the expressway, it needs to continuously pay attention to the driving situation of the surrounding target vehicles, especially for the maneuvering targets adjacent to the target. There may be two or more measurements associated with this; at this time, the target no longer satisfies the one-to-one assumption (one-to-one correspondence between targets and measurements), and even the increase in the number of targets and measurements leads to missed or false detections [27]. Therefore, the research plan needs to solve the problems of many measurement interference sources in complex driving scenarios, which are prone to misidentification or missed detection.

Considering the autonomous-vehicle sensor configuration, sensor characteristics, and environmental factors, millimeter-wave radar and lidar were selected for information fusion to detect and track target characteristics. The autonomous vehicle was equipped with four millimeter-wave radars and a 32-line lidar. Among them, the millimeter-wave radars were respectively arranged on the front, rear, left, and right sides of the vehicle body, and the range and azimuth resolutions of the radars were 2.5 m and 6°, respectively. The millimeter-wave radar installed directly in front of the vehicle was a long-range radar with a detection range of 250 m and a horizontal field of view of 30°; the millimeter-wave radar equipped directly behind the vehicle was a medium-range radar with a detection range of 100 m and a horizontal field of view of 90°; and the millimeter-wave radars equipped on the left and right sides of the vehicle body were short-range radars with a detection range of 30 m and a horizontal field of view of 160°. In addition, the lidar was configured on the top of the vehicle body; the detection distance, horizontal field of view, and vertical field of view were 200 m, 360°, and 40°, respectively, and the distance resolution, horizontal field of view resolution, and vertical field of view resolution were 5 cm, 0.2°, and 0.4° respectively.

According to the needs of autonomous vehicles and sensor selection, this paper proposes a multitarget-tracking method for autonomous vehicles that comprehensively considers key technologies such as target tracking, sensor registration, track association, and data fusion. The millimeter-wave radar and lidar sensors first formed their own target tracks based on the single-sensor multitarget-tracking algorithm, and then sent the target tracks to the fusion center to complete the space–time registration and track association; then, the fusion center estimated and fused the tracks from the same target based on the estimated fusion criteria, and finally formed a stable and high-precision global track. The overall framework is shown in Figure 1, and the data fusion in the figure only represents the final stage of multisensor information fusion.

The specific steps are as follows:Single-sensor target tracking: The extended target tracker is constructed according to the GM-PHD algorithm and the rectangular target model, and the millimeter-wave radar sensor uses the extended target tracker to track multiple targets and generate the local track of the target object. Then, a JPDA tracker configured with the Interacting multiple modules–unscented Kalman filter (IMM-UKF) algorithm is built, which is used by the lidar to track multiple targets and generate the local tracks of target objects.The spatiotemporal registration of sensors: The Kalman filter method is used to register the asynchronous measurement information of each sensor to the same target at the same time, so as to realize sensor time registration. The Residual Bias Estimation Registration (RBER) method is used to estimate and compensate the detection information of the space public target, so as to realize sensor space registration.Sensor track association: using the sequential m-best track association algorithm based on the new target density (SMBTANTD), each time in an iterative manner, tracks from the next sensor are introduced and correlated with the previous results.Sensor data fusion: the IF heterogeneous sensor fusion algorithm is used to avoid the repeated calculation of public information and realize the optimal combination of track information provided by millimeter-wave radar and lidar, so as to obtain more accurate target status information.

## 3. Track Fusion and Management

### 3.1. Single-Sensor Target Tracking

The essence of target tracking is the process of estimating the state and number of targets by using the noisy measurement data obtained by each sensor [28]. Considering the configuration and characteristics of the millimeter-wave radar sensor and the lidar sensor comprehensively, the corresponding target-tracking research was carried out.

#### 3.1.1. Millimeter-Wave Radar Target Tracking

In traditional tracking methods such as Global Nearest Neighbor (multiObjectTracker and trackerGNN), Joint Probabilistic Data Association (trackerJPDA), and Multiple Hypothesis Tracking (trackerTOMHT), the tracked object is considered to satisfy the one-to-one hypothesis that each scan of the sensor will return a detected target value. However, for a millimeter-wave radar mounted on an autonomous vehicle, due to the improvement in its resolution, there may be two or more measurements associated with the measured target. In addition, for autonomous vehicles driving on highways, most of the measured targets are also high-speed vehicles. In view of this, the GM-PHD algorithm and rectangular target model were used to construct an extended target tracker.

According to the rectangular target model of the measured object, the rectangular extended target state of the measured object was obtained; the rectangular extended target state is expressed as:(1)$$\xi =\left(\gamma ,x,X\right),$$
among them, *ξ* represents the state vector of the extended target of the detected object; *γ* represents the measurement rate state of the extended target of the measured object, which is a scalar subject to the gamma distribution; ***x*** represents the motion state of the extended target of the measured object; the vector ***x*** can be modeled as ***x*** = [*x*, *y*, *v*, *θ*, *ω*, *L*, *W*]*^T^*; [*x*, *y*]*^T^* represents the two-dimensional coordinate position information of the measured target; *v*, *θ*, and *ω* represent the speed, direction angle, and angular velocity of the target, respectively; *L* and *W* represent the length and width of the detection target, respectively; and ***X*** represents the extended state of the extended target of the measured object, that is, the state after the expansion of a single piece of measurement information. When the extended target measurement is distributed in a certain space in disorder, usually a rectangle can be used to approximate the extended state of the extended target.

We constructed the GM-PHD filter according to the rectangular extended target state of the detected object; the GM-PHD filter approximates the multitarget probability hypothesis density through the Gaussian component with weights, assuming that the multitarget posterior PHD at *k*−1 time can be expressed as a Gaussian mixture:(2)$$D_{k-1}\left(x\right)=\sum_{i=1}^{J_{k-1}}\omega_{k-1}^{i}N\left(x;m_{k-1}^{i},P_{k-1}^{i}\right),$$
among them, *J_k_*_−1_ is the number of Gaussian components at time *k*−1, $\omega_{k-1}^{i}$ is the weight of the *i*-th Gaussian component, *N*(·; *m*, *P*) represents the probability density function of Gaussian distribution, *N*(***x***; $m_{k-1}^{i}$, $P_{k-1}^{i}$) is the probability density function of the *i*-th Gaussian distribution at time *k*−1, the mean is $m_{k-1}^{i}$, and the covariance is $P_{k-1}^{i}$.

Multitarget prediction PHD at time *k* and multitarget posterior PHD at time *k* can also be expressed in Gaussian mixture form:(3)$$D_{k\left|k-1\right.}\left(x\right)=D_{S,k\left|k-1\right.}\left(x\right)+\gamma_{k}\left(x\right)=\sum_{i=1}^{J_{k\left|k-1\right.}}\omega_{k\left|k-1\right.}^{i}N\left(x;m_{k\left|k-1\right.}^{i},P_{k\left|k-1\right.}^{i}\right),$$
(4)$$D_{k\left|k\right.}\left(x\right)=\left(1-p_{D,k}\right)D_{k\left|k-1\right.}\left(x\right)+\sum_{Z\in Z_{k}}\sum_{j=1}^{J_{k\left|k-1\right.}}\omega_{k}^{j}\left(z\right)N\left(x;m_{k\left|k\right.}^{j}\left(z\right),P_{k\left|k\right.}^{j}\right),$$
among them, ***x*** represents the motion state of the rectangular extended target state of the detected object; *D*_*S*,*k*|*k*−1_(***x***) represents the multitarget prediction PHD at time *k*; *D*_*S*,*k*|*k*−1_(***x***) represents the PHD of the surviving Gaussian component at time *k*; *γ_k_*(***x***) represents the new target at time *k*, that is, the new observation point PHD acquired by the sensor; $\omega_{k\left|k-1\right.}^{i}$ represents the weight of the *i*-th Gaussian component at time *k*; *N*(***x***; $m_{k\left|k-1\right.}^{i}$, $P_{k\left|k-1\right.}^{i}$) means the Gaussian component whose mean is $m_{k\left|k-1\right.}^{i}$ and whose covariance is $P_{k\left|k-1\right.}^{i}$; *J_k|k_*_−1_ means the number of Gaussian components predicted at time *k*; *D_k_*_|*k*_(***x***) represents the posterior multitarget PHD at time *k*; *p*_*D*,*k*_ represents the target detection probability, which means that when there is a signal at the input end of the millimeter-wave radar, due to noise, two judgment results may be concluded—that there is a “signal” or “no signal”—and when there is a signal, it will provide the probability that millimeter-wave radar will make a correct decision of “signal”; $\omega_{k}^{i}$(*z*) represents the weight of the *j*-th update component; *z_k_* represents the measured value of the target at time *k*, which refers to the physical information value such as the target coordinates obtained by the millimeter-wave radar; *N*(***x***; $m_{k|k}^{j}$(*z*), $P_{k|k}^{j}$) means that the mean is $m_{k|k}^{j}$(*z*); and the covariance is the Gaussian component of $P_{k|k}^{j}$. The GM-PHD filter was constructed by predicting *D_k_*_|*k*−1_(***x***) and updating *D_k_*_|*k*_(***x***) through the GM-PHD algorithm, and then the track tracker suitable for millimeter-wave radar was obtained, that is, the extended target tracker.

#### 3.1.2. LiDAR Target Tracking

The amount of measurement information obtained by lidar measurements for each target is far more than that of a millimeter-wave radar; it will also obtain ground measurement information, and the amount of data output by it is not the same order of magnitude as that of a millimeter-wave radar. If the extended object tracker is directly used for processing, the output signal will seriously lag due to the high computational complexity. Therefore, it is necessary to construct a bounding box detector to reduce the dimensionality of the 3D detection information of the lidar.

When constructing a bounding box detector, a plane fitting algorithm based on Random Sample Consensus (RANSAC) [29] is used to preprocess the 3D data of the lidar to remove redundant point cloud information such as the ground, and then obtain the target point cloud, reducing the computational overhead. The specific process is as follows: first, randomly select three points in the initial point cloud, and calculate the sampling plane formed by the three points; secondly, calculate the distance from all point clouds to the sampling plane, set the distance threshold, and divide many point clouds into inner points (normal data) and outer points (abnormal data) according to the threshold; then, count the number of interior points and exterior points, repeat the above process to the maximum number of iterations, and select the plane with the most interior points; and finally, based on the plane with the most interior points, refit the plane with all its interior points to obtain the final fitting plane equation, remove redundant point clouds such as ground points according to the final fitting plane equation, and obtain the target point cloud.

After removing redundant point clouds, point clouds belonging to various main objects will be presented in a state of floating, separated in space, and cuboid bounding box detectors will be used to detect objects. First, the Euclidean algorithm is used to cluster the target point cloud in this state, and the state vector of the bounding box detector is obtained according to the clustering result. The state vector of the bounding box detector is expressed as:(5)$$x^{\prime }={\left[x,y,v,\theta ,\omega ,z,\dot{z},L,W,H\right]}^{T},$$
among them, ***x***′ represents the state vector, compared with the state vector of the rectangular model of the millimeter-wave radar; the state vector of the cuboid bounding box detector has three more variables: *z*, $\dot{z}$, and *H*. *z* represents the vertical coordinate of the detection target, $\dot{z}$ represents the detection target’s vertical velocity, and *H* represents the height of the detection target.

Since a single motion model is not effective for describing maneuvering targets, an interactive multimodel unscented Kalman filter (IMM-UKF) was used in this paper to estimate and update the state. The IMM algorithm can synthesize multiple moving targets [30] and analyze the weights of each model through Bayesian theory, which can more accurately describe the law of vehicle motion. The basic idea is to use a finite number of models to describe the motion state of the target as comprehensively as possible; the transition between the models is subject to the Markov state chain, and the interaction between multiple models is considered. The JPDA tracker configured with IMM-UKF is a point target tracker, as shown in Figure 2. The tracker consists of an input interaction module, a UKF filter module, a probability update module, a JPDA data association module, and an output fusion module; it is used to achieve local track generation for objects detected by lidar.

The UKF filter obtains the first state estimate ${\hat{x}}_{j}^{i}\left(k\left|k\right.\right)$ based on the state vector ***x***ʹ of the bounding box detector at time *k*.

The input interaction module calculates the second state estimate ${\hat{x}}_{j}^{0i}\left(k\left|k\right.\right)$ and the second covariance matrix $P_{j}^{0i}\left(k\left|k\right.\right)$ after multiple target interactions according to the first state estimate ${\hat{x}}_{j}^{i}\left(k\left|k\right.\right)$ and the first covariance matrix $P_{j}^{i}\left(k\left|k\right.\right)$ of the UKF filter in the UKF filter module at time *k* and outputs, where *j* represents the motion model, which is a velocity model constructed according to the motion state of the target, *i* = 1, 2, …, *r*; *r* is the number of UKF filters.
(6)$${\hat{x}}_{j}^{0i}\left(k\left|k\right.\right)=\sum_{i=1}^{N}{\hat{x}}_{j}^{i}\left(k\left|k\right.\right)\mu_{j\left|i\right.}\left(k\right),$$
(7)$$P_{j}^{0i}\left(k\left|k\right.\right)=\sum_{i=1}^{N}\left\{P_{i}\left(k\left|k\right.\right)+\left[{\hat{x}}_{j}^{i}\left(k\left|k\right.\right)-{\hat{x}}_{j}^{0i}\left(k\left|k\right.\right)\right]∙{\left[{\hat{x}}_{j}^{i}\left(k\left|k\right.\right)-{\hat{x}}_{j}^{0i}\left(k\left|k\right.\right)\right]}^{T}\right\}\mu_{j\left|i\right.}\left(k\right),$$
in Formulas (6) and (7), ${\hat{x}}_{i}\left(k\left|k\right.\right)$ is the state estimation of the target at time *k* under model *i*, $P_{i}\left(k\left|k\right.\right)$ is the estimated variance of the target at time *k* under model *i*, and $\mu_{j\left|i\right.}\left(k\right)$ is the interactive input probability.

The UKF filter in the UKF filter module runs the UKF filter in parallel through the time update process described by Formula (8) and the measurement update process described by Formula (9) according to the output of the input interaction module and the effective observation vector *Z_k_* at time *k*, then outputting the third state estimate ${\hat{x}}_{j}^{i}\left(k+1\left|k\right.\right)$, and the third covariance matrix $P_{j}^{i}\left(k+1\left|k\right.\right)$ at time *k* + 1 respectively.
(8)$${\hat{x}}_{j}^{i}\left(k+1\left|k\right.\right)=F_{j}\left(k\right)\;{\hat{x}}_{j}^{0i}\left(k\left|k\right.\right),$$
(9)$$P_{j}^{i}\left(k+1\left|k\right.\right)=F_{j}\left(k\right)P_{j}^{0i}\left(k\left|k\right.\right){\left[F_{j}\left(k\right)\right]}^{T}+Q_{j}\left(k\right),$$
among them, ***F****_j_*(*k*) is the state transition matrix of model *j* at time *k*; ***Q****_j_*(*k*) is the process noise of model *j* at time *k.*

Then, the information of the *i*-th measurement of the target under model *j* is as follows:(10)$$v_{j}^{\left(i\right)}\left(k+1\right)=z^{\left(i\right)}\left(k+1\right)-H_{j}\left(k+1\right){\hat{x}}_{j}^{i}\left(k+1\left|k\right.\right),$$
its covariance matrix ***S****_j_*(*k* + 1) is as follows:(11)$$S_{j}\left(k+1\right)=H_{j}\left(k+1\right)P_{j}^{i}\left(k+1\left|k\right.\right){\left[H_{j}\left(k+1\right)\right]}^{T}+R_{j}\left(k+1\right),$$
among them, ***z***^(*i*)^(*k* + 1) is the *i*-th observation value associated with the target at time *k* + 1; ***H****_j_*(*k* + 1) is the observation matrix of model *j* at time *k* + 1; and ***R****_j_*(*k* + 1) is the noise covariance matrix.

The probability update module calculates the conditional probability $\mu_{j}^{i}\left(k+1\right)$ of the motion model *j* at time *k* + 1 according to the residual information of the UKF filter module, where the residual information of the UKF filter is denoted by $\Lambda_{j}^{i}\left(k+1\right)$.
(12)$$\mu_{j}\left(k+1\right)=\frac{\mu_{j}\left(k+1\left|k\right.\right)\Lambda_{j}^{i}\left(k+1\right)}{\sum_{i=1}^{N}\mu_{j}^{i}\left(k+1\left|k\right.\right)\Lambda_{j}^{i}\left(k+1\right)},$$
the JPDA data association module calculates the fusion information under the model *j* at the time *k* + 1 of the target according to Formula (13):(13)$${\tilde{v}}_{j}\left(k+1\right)=\sum_{i=1}^{M}\beta_{j}^{\left(i\right)}v_{j}^{\left(i\right)}\left(k+1\right),$$
among them, *M* is the number of observations currently associated; $\beta_{j}^{\left(i\right)}$ is the probability of association with the target *k* + 1 time and the *i*-th observation under model *j*.

The Kalman gain matrix can be expressed as:(14)$$K_{j}\left(k+1\right)=P_{j}^{i}\left(k+1\left|k\right.\right){\left[H_{j}\left(k+1\right)\right]}^{T}∙{\left[S_{j}\left(k+1\right)\right]}^{-1},$$
updating the state variance, we can obtain:(15)$$\begin{array}{ll} P_{j}\left(k+1\left|k+1\right.\right)= & P_{j}^{i}\left(k+1\left|k\right.\right)-\left(\sum_{i=1}^{M}\beta_{j}^{\left(i\right)}\right)∙K_{j}\left(k+1\right)S_{j}\left(k+1\right){\left[K_{j}\left(k+1\right)\right]}^{T}+\\ & K_{j}\left(k+1\right)\left\{\sum_{i=1}^{M}\beta_{j}^{\left(i\right)}v_{j}^{\left(i\right)}\left(k+1\right){\left[v_{j}^{\left(i\right)}\left(k+1\right)\right]}^{T}-{\left[{\tilde{v}}_{j}\left(k+1\right)\right]}^{T}\right\}{\left[K_{j}\left(k+1\right)\right]}^{T} \end{array}$$

The output fusion module calculates the fused state estimation and covariance matrix according to the conditional probability $\mu_{j}\left(k+1\right)$ of motion model *j* at time *k* + 1, and obtains the fused state estimation $\hat{x}\left(k+1\left|k+1\right.\right)$ and covariance matrix ***P***(*k* + 1|*k* + 1).
(16)$$\hat{x}\left(k+1\left|k+1\right.\right)=\sum_{j=1}^{N}{\hat{x}}_{j}\left(k+1\left|k+1\right.\right)∙\mu_{j}\left(k+1\right),$$
(17)$$P\left(k+1\left|k+1\right.\right)=\sum_{j=1}^{N}\mu_{j}\left(k+1\right)\left[P_{j}\left(k+1\left|k+1\right.\right)+\Delta {\hat{x}}_{j}∙\Delta {\hat{x}}_{j}{}^{T}\right],$$
among them, $\Delta {\hat{x}}_{j}={\hat{x}}_{j}\left(k+1\left|k+1\right.\right)-\hat{x}\left(k+1\left|k+1\right.\right)$.

### 3.2. Sensor Spatiotemporal Registration

The measurement information of the millimeter-wave radar and the laser radar are respectively processed by their respective local trackers to form two local tracks, that is, the millimeter-wave radar track information of the detected object and the millimeter-wave radar track information of the detected object. Due to the different transmission rates and sampling periods of different sensors and there being sensor system deviations and measurement errors, direct conversion will reduce the accuracy of data fusion. Therefore, sensor spatiotemporal registration is required when processing millimeter-wave radar and lidar data.

#### 3.2.1. Time Registration

Time registration is to register the asynchronous measurement information of each sensor to the same target at the same time [31]. Typical time registration methods include the least squares method [32], curve fitting method [33], and Kalman filter method [30]. The least squares method is only suitable for uniform moving objects, and it will cause model mismatch for nonuniform moving objects, resulting in unsatisfactory registration results. The curve fitting and Kalman filtering methods are suitable for uniform and nonuniform sampling scenarios. The Kalman filtering method can adjust the target motion model. Compared with the curve fitting method, the registration accuracy is higher when the moving target is complex. Therefore, in this paper, a Kalman filter-like method was used for the temporal registration of sensor fusion. In the highway driving scenario, the tracked target may maneuver, and a single model is not enough to describe its state. To achieve temporal registration in a maneuvering target-tracking system, the IMM-TF method [34] configured with a Kalman filter algorithm was adopted to solve the temporal offset estimation problem.

For a multisensor system, the relative time offset between sensors *i* and *n* at time k is defined as $\Delta {\hat{t}}_{k}^{i,n}$. First, a two-stage relative time offset estimation algorithm is used to calculate the relative time offset estimation between sensor 1 and other sensors, such as $\Delta {\hat{t}}_{k}^{1,2}$, $\Delta {\hat{t}}_{k}^{1,3}$, …, $\Delta {\hat{t}}_{k}^{1,N_{r}}$. Secondly, the global accurate time stamp is obtained using Formula (18).
(18)$$N=\left\{\begin{array}{l} 1,if\;\underset{n}{\max}\left\{\Delta {\hat{t}}_{k}^{i,n},n=2,3,\cdots ,N_{r}\right\}\leq 0\\ \underset{n}{\arg\max}\left\{\Delta {\hat{t}}_{k}^{i,n},n=2,3,\cdots ,N_{r}\right\},\mathrm{otherwise}. \end{array}\right.,$$Therefore, the relative time offset between sensor *i* and the global precise timestamp is as follows:(19)$$\Delta {\hat{t}}_{k}^{i,N}=\Delta {\hat{t}}_{k}^{1,i}-\Delta {\hat{t}}_{k}^{1,N},$$
and the unbiased measurement coordinate system conversion algorithm is used to convert the polar coordinate measurement into Cartesian coordinates. The conversion measurement equation of sensor *i* at time stamp $t_{k}^{-i}$ is as follows:(20)$$Z_{i}\left({\bar{t}}_{k}^{i}\right)=H\left(t_{k}^{i}\right)x\left(t_{k}^{i}\right)+v_{i}\left(t_{k}^{i}\right),$$
among them, $H\left(t_{k}^{i}\right)$ is the linear measurement transfer matrix, $x\left(t_{k}^{i}\right)$ is the passive dynamic equation, and $v_{i}\left(t_{k}^{i}\right)$ is the zero-mean white Gaussian noise vector. Due to the time offset, the target dynamic equation that needs to be filtered is as follows:(21)$$x\left({\bar{t}}_{k+1}^{i}\right)=F\left({\bar{t}}_{k+1}^{i}\left|{\bar{t}}_{k}^{i}\right.\right)x\left({\bar{t}}_{k}^{i}\right)+w\left({\bar{t}}_{k+1}^{i}\left|{\bar{t}}_{k}^{i}\right.\right),$$
and among them, $F\left({\bar{t}}_{k+1}^{i}\left|{\bar{t}}_{k}^{i}\right.\right)$ is the state transition matrix from time ${\bar{t}}_{k}^{i}$ to time ${\bar{t}}_{k+1}^{i}$, and $w\left({\bar{t}}_{k+1}^{i}\left|{\bar{t}}_{k}^{i}\right.\right)$ is the zero-mean white Gaussian noise of the covariance matrix.

For different target dynamic models, such as CV, CA, or CT models, the corresponding estimators are used to calculate constant or time-varying relative time offset estimates. For the calculation methods, refer to [34].

Finally, the time offset estimates of different dynamic models are fused, and the obtained time offset estimates and covariance are as follows:(22)$$\Delta {\hat{t}}_{k}^{1,2}=\sum_{j}\Delta {\hat{t}}_{k}^{j,1,2}\mu^{j}\left({\bar{t}}_{k}^{i}\right),$$
(23)$$\sum_{k}^{1,2}=\sum_{j}\left\{\sum_{k}^{j,1,2}+\left[\Delta {\hat{t}}_{k}^{1,2}-\Delta {\hat{t}}_{k}^{j,1,2}\right]∙{\left[\Delta {\hat{t}}_{k}^{1,2}-\Delta {\hat{t}}_{k}^{j,1,2}\right]}^{T}\right\}\mu^{j}\left({\bar{t}}_{k}^{i}\right),$$
and among them, $\mu^{j}\left({\bar{t}}_{k}^{i}\right),$ is the conditional probability of motion model *j*.

#### 3.2.2. Spatial Registration

Spatial registration is the process of using multisensors to detect the information of common space targets and estimating and compensating the system deviation of sensors, which can improve the accuracy of information fusion [35]. Lidar can obtain the complete three-dimensional position information of the target (distance R, pitch angle θ, and azimuth angle φ), while millimeter-wave radar cannot obtain complete three-dimensional information, and can only provide angle and distance information. Therefore, in order to deal with the problem of incomplete measurement data registration model mismatch, this paper uses the Residual Bias Estimation Registration (RBER) method [36] to estimate and compensate for the detection information of space public targets.

All measurement values are divided into two parts: complete measurement and noncomplete measurement. First, a maximum likelihood estimation is performed on the complete measurements in a common coordinate system to find the target position estimate ${\hat{x}}_{k}^{c}$:(24)$$\left\{\begin{array}{l} {\hat{x}}_{k}^{c}=\sum_{{\hat{x}}_{k}}\left(\sum_{i=1}^{N_{C}}\sum_{{\hat{x}}_{i,k}}^{-1}{\tilde{x}}_{i,k}\right)\\ \sum_{{\hat{x}}_{k}}={\left(\sum_{i=1}^{N_{C}}\sum_{{\hat{x}}_{i,k}}^{-1}\right)}^{-1} \end{array}\right.,$$

Then, based on sequential filtering technology, the incomplete measurement data are used to sequentially update ${\hat{x}}_{k}^{c}$ to obtain the updated target position estimate ${\hat{x}}_{k}^{all}$.
(25)$$\left\{\begin{array}{l} {\hat{x}}_{k}^{all}={\hat{x}}_{k}^{c}+K_{k}\left[z_{k}^{In}-h\left({\hat{x}}_{k}^{c}\right)\right]\\ K_{k}=P_{k}^{all}{\left(H_{x}^{In}\right)}^{T}{\left(\sum_{z_{k}}^{In}\right)}^{-1}\\ P_{k}^{all}={\left(P_{k}^{c}\right)}^{-1}+{\left(H_{x}^{In}\right)}^{T}{\left(\sum_{z_{k}}^{In}\right)}^{-1}H_{x}^{In} \end{array}\right.,$$
and among them, ${\hat{x}}_{k}^{c}$ and $P_{k}^{c}$ are the complete measurement target position estimation and covariance matrix estimation, respectively; $H_{x}^{In}$ is the connection Jacobian matrix about *L* sensors; and $\sum_{z_{k}}^{In}$ is the covariance matrix composed of incomplete measurement data.

Then, ${\hat{x}}_{k}^{all}$ is converted to the measurement coordinate system, a maximum likelihood estimation is performed for all measurements in the measurement coordinate system, the parameter *ρ* is obtained to be estimated through iteration, and the measurement information of the salient target is used to eliminate the systematic deviation of the sensor. The specific calculation method can be found in reference [36].

### 3.3. Sensor Track Association

In a distributed multisensor multitarget tracking system, the purpose of track association is to study how to match the tracks reported by different sensors. Due to the asynchrony of the track and the existence of systematic errors [37], the difficulty of track correlation is increased, and the above problems must be solved in order to obtain reliable correlation results. In order to solve this problem, a sequential m-best track association algorithm based on the new target density (SMBTANTD) [36] is sampled. This method introduces the history of track information to improve track correlation performance. In addition, this method can effectively solve the coupling problem between track association and spatial registration by utilizing the repeated process of salient target selection, multitarget filter tracking, and different sensor target association [36].

After each sensor obtains the target measurement value, the measurement value is filtered to obtain the target state estimation value $\hat{x}$ and its error covariance matrix ***P***. If $\Gamma_{j}^{i}$ represents the tracking result of sensor *i* on target *j*, then the tuple,
(26)$$\Upsilon_{j}=\left\{\Gamma_{j}^{i}\left|i=1,2,\cdots ,M\right.\right\},$$
represents the hypothesis tracked from the same target.

In a multitarget multisensor, when creating a global association hypothesis *H* consisting of tuples, the goal is to find the most probable hypothesis in the set of all global hypotheses. The likelihood function of a tuple can be expressed as:(27)$$L\left(\Upsilon_{j}\right)=p\left(\Gamma_{j}^{i},\Gamma_{j}^{i},\cdots ,\Gamma_{j}^{M}\left|\Upsilon_{j}\right.\right),$$
and the most likely global hypothesis can be obtained by finding the tuple with the largest likelihood function [38], namely:(28)$$\hat{H}=\arg\max\prod_{\Upsilon \in H}L\left(\Upsilon \right),$$Assuming that multiple trajectories representing the same target obey a zero-mean normal distribution with variance *P*, there is no correlation between trajectories. The likelihood function can be expressed as:(29)$$L\left(\Upsilon \right)=\prod_{i=1}^{M}\frac{1}{{\left|2\pi P_{if}\right|}^{1/2}}\times \exp\left[-\frac{1}{2}{\left({\hat{x}}_{i}-{\hat{x}}_{F,M}\right)}^{T}P_{if}^{-1}\left({\hat{x}}_{i}-{\hat{x}}_{F,M}\right)\right],$$
and among them, $P_{if}$ represents the cross-covariance matrix of random variables, and ${\hat{x}}_{i}$ represents the state estimation of the *i*-th sensor. ${\hat{x}}_{F,M}$ represents the global state estimate.
(30)$$P_{if}=P_{ii}-2P_{F,M}+P_{F,M}\left(P_{ii}^{-1}+P_{F,M-1}^{-1}\right)P_{F,M},$$
and among them, $P_{ii}$ is the cross-covariance matrix.

Let *c_ji_* denote the *j*-th row and *i*-th column of the matrix; then, the target allocation cost among different sensors can be expressed as:(31)$$c_{ji}=-\ln L\left(\Upsilon \right),$$
and among them, ln *L*(Υ) does not depend on the target indices *i* and *j*; it depends on different combinations of different target trajectories from different sensors.

In order to solve the correlation problem when the number of targets measured by multiple sensors is inconsistent, a new target density is introduced in the correlation matrix. The target j acquired from the next sensor is defined as the new target, namely:(32)$$c_{j}=-\ln\left(\rho_{N}\right),$$
where $\rho_{N},$ is the density of new objects, which has similarity with the correlation threshold [39].

Assuming that object trajectories from the same sensor are not correlated, the allocation cost is set to infinity. Based on the Murty algorithm [40], the optimal solution to the two-dimensional assignment problem is obtained:(33)$$\left\{\begin{array}{l} \Upsilon_{2}=\left\{\Upsilon_{2,1},\Upsilon_{2,2},\Upsilon_{2,3}\right\}\\ \Upsilon_{2,1}=\left\{\Gamma_{1}^{1},\Gamma_{1}^{2}\right\}=\left\{\Gamma_{2,2}^{2,1}\right\}\\ \Upsilon_{2,2}=\left\{\Gamma_{2}^{1},\Gamma_{2}^{2}\right\}=\left\{\Gamma_{2,2}^{2,1}\right\}\\ \Upsilon_{2,3}=\left\{\Gamma_{3}^{1}\right\} \end{array}\right.,$$
and among them, $\Gamma_{1}^{1}$, $\Gamma_{2}^{1}$, and $\Gamma_{3}^{1}$ represent the track information of sensor 1 for three targets; $\Gamma_{1}^{2}$ and $\Gamma_{2}^{2}$ represent the track information of sensor 2 for two targets; Υ_2,*i*_ is the *i*-th tuple; and $\Gamma_{i}^{j}$ represents the state estimation of the *j*-th sensor to *i* targets and the tracking track information of the covariance. Υ_2,1_ = {$\Gamma_{1}^{1}$, $\Gamma_{1}^{2}$} means the matching information of the first target received by sensor 1 and the first target received by sensor 2. Υ_2,2_ = {$\Gamma_{2}^{1}$, $\Gamma_{2}^{2}$} means that the information of the second target received by sensor 1 matches the information of the first target received by sensor 2. Υ_2,3_ = {$\Gamma_{3}^{1}$} means that the third target information received by sensor 1 does not match.

Finally, each time in an iterative manner, tracks from the next sensor are introduced and correlated with the previous results.

### 3.4. Heterogeneous Track-to-Track Fusion

Millimeter-wave radar and lidar are sensors of different configurations. Heterogeneous sensor data fusion can use the incomplete measurement information in the measurement coordinate system to update the complete target state estimation in the Cartesian coordinate system [41]. In addition, the system is a nonlinear system; since the closed-form solution of the posterior probability density function is very difficult, this paper adopts the nonlinear system suboptimal filtering algorithm, that is, the IF heterogeneous sensor fusion algorithm [42] completes the fusion of millimeter-wave radar and lidar measurement data.

${\hat{x}}_{k|k}^{i}$ and $P_{k|k}^{i}$ denote the state estimation of the *i*-th tracker at the moment $t_{k}^{i}$ and the corresponding covariance of the Cartesian coordinate system, respectively, since two different trackers are used for millimeter-wave radar and lidar, respectively (let them be tracker 1 and tracker 2); so, here, *i* = 1, 2. For positive integers *j* and *l*, their information matrix and information state are estimated as:(34)$$\Upsilon_{j|l}:=P_{j|l}^{-1},$$
(35)$${\hat{\gamma }}_{j|l}:=P_{j|l}^{-1}{\hat{x}}_{j|l}^{-1}=\Upsilon_{j|l}{\hat{x}}_{j|l},$$

The starting moment of fusion is $t_{k_{1}}^{1}=t_{j_{1}}^{2}$, $\left({\hat{x}}_{k_{1}|k_{1}}^{1},P_{k_{1}|k_{1}}^{1}\right)$ and $\left({\hat{x}}_{j_{1}|j_{1}}^{2},P_{j_{1}|j_{1}}^{2}\right)$ are obtained from tracker 1 and tracker 2, respectively, and the state estimation and covariance in the Cartesian coordinate system are transformed into a corresponding information state estimation and information matrix.
(36)$$\Upsilon_{k_{1}|k_{1}}^{1}={\left(P_{k_{1}|k_{1}}^{-1}\right)}^{-1},{\hat{\gamma }}_{k_{1}|k_{1}}^{1}=\Upsilon_{k_{1}|k_{1}}^{1}{\hat{x}}_{k_{1}|k_{1}}^{1},$$
(37)$$\Upsilon_{j_{1}|j_{1}}^{2}={\left(P_{j_{1}|j_{1}}^{2}\right)}^{-1},{\hat{\gamma }}_{j_{1}|j_{1}}^{2}=\Upsilon_{j_{1}|j_{1}}^{2}{\hat{x}}_{j_{1}|j_{1}}^{2},$$

At time $t_{k_{1}}^{1}$, tracker 1 and tracker 2 send $\left(\Upsilon_{k_{1}|k_{1}}^{1},{\hat{\gamma }}_{k_{1}|k_{1}}^{-1}\right)$ and $\left(\Upsilon_{j_{1}|j_{1}}^{2},{\hat{\gamma }}_{j_{1}|j_{1}}^{2}\right)$ to the fusion center, respectively. Make $t_{k_{1}}=t_{k_{1}}^{1}$, $\Upsilon^{1}\left(t_{k_{1}}\right)=\Upsilon_{k_{1}|k_{1}}^{1}$, and $\Upsilon^{2}\left(t_{k_{1}}\right)=\Upsilon_{j_{1}|j_{1}}^{2}$. By analogy, the fusion information matrix and the corresponding state matrix at time $t_{k_{1}}$ of the calculation by the fusion center are as follows:(38)$$\Upsilon \left(t_{k_{1}}\right)=\Upsilon^{1}\left(t_{k_{1}}\right)+\Upsilon^{2}\left(t_{k_{1}}\right),\hat{\gamma }\left(t_{k_{1}}\right)={\hat{\gamma }}^{1}\left(t_{k_{1}}\right)+{\hat{\gamma }}^{2}\left(t_{k_{1}}\right),$$Let $t_{cf}$ represent the current fusion moment and set $t_{cf}=t_{k_{1}}$; then, the time points of the data fusion center are $t_{k_{1}}$, $t_{k_{1}}+N_{f}\Delta$, …, $t_{f}$ in turn. Among them, Δ is a constant data measurement interval, *N*_f_ is a positive integer, and *t*_f_ is the final time.

Let $t_{k}=t_{cf}+N_{f}\Delta$; the tracker sends information to the fusion center at time $t_{k}$. For local tracker 1, the prediction information matrix and the corresponding state estimation from time $t_{cf}$ to time $t_{k}$ are as follows:(39)$$\Upsilon^{1}\left(t_{k},t_{cf}\right)=\left[F\left(t_{k},t_{cf}\right){\left(\Upsilon^{1}\left(t_{cf}\right)\right)}^{-1}F^{\prime }\left(t_{k},t_{cf}\right)+Q\left(t_{k},t_{cf}\right)\right],$$
(40)$${\hat{\gamma }}^{1}\left(t_{k},t_{cf}\right)=\Upsilon^{1}\left(t_{k},t_{cf}\right)F\left(t_{k},t_{cf}\right){\left(\Upsilon^{1}\left(t_{k},t_{cf}\right)\right)}^{-1}{\hat{\gamma }}^{1}\left(t_{cf}\right),$$
among them, $F\left(t_{k},t_{cf}\right)$ and $Q\left(t_{k},t_{cf}\right)$ are the state transition matrix and process noise covariance matrix from time $t_{cf}$ to time $t_{k}$, respectively. For local tracker 2, a similar method can be used to obtain the prediction information matrix $\Upsilon^{2}\left(t_{k},t_{cf}\right)$ from time $t_{cf}$ to time $t_{k}$ and the corresponding state estimation ${\hat{\gamma }}^{2}\left(t_{k},t_{cf}\right)$. Furthermore, the information matrix $\Upsilon \left(t_{k},t_{cf}\right)$ and state estimation $\hat{\gamma }\left(t_{k},t_{cf}\right)$ of the fusion center can be obtained.

When new information is sent to the fusion center, the information state matrix and information matrix of time $t_{k}$ tracker 1 and tracker 2 need to be calculated.
(41)$$\Upsilon^{1}\left(t_{k}\right)={\left(P^{1}\left(t_{k}\right)\right)}^{-1},{\hat{\gamma }}^{1}\left(t_{k}\right)=\Upsilon^{1}\left(t_{k}\right){\hat{x}}^{1}\left(t_{k}\right),$$
(42)$$\Upsilon^{2}\left(t_{k}\right)={\left(P^{2}\left(t_{k}\right)\right)}^{-1},{\hat{\gamma }}^{2}\left(t_{k}\right)=\Upsilon^{2}\left(t_{k}\right){\hat{x}}^{2}\left(t_{k}\right),$$

The new information matrix and information state estimate sent by tracker 1 to the fusion center at time $t_{k}$ are as follows:(43)$$\Delta \Upsilon^{1}\left(t_{k}\right)=\Upsilon^{1}\left(t_{k}\right)-\Upsilon^{1}\left(t_{k},t_{cf}\right),$$
(44)$$\Delta {\hat{\gamma }}^{1}\left(t_{k}\right)={\hat{\gamma }}^{1}\left(t_{k}\right)-{\hat{\gamma }}^{1}\left(t_{k},t_{cf}\right),$$

For local tracker 2, a similar method can be used to obtain the new information matrix $\Delta \Upsilon^{2}\left(t_{k}\right)$ and information state estimation $\Delta {\hat{\gamma }}^{2}\left(t_{k}\right)$ sent to the fusion center from time $t_{k}$.

Based on the predicted information matrix $\Upsilon \left(t_{k},t_{cf}\right)$ and information state estimation $\hat{\gamma }\left(t_{k},t_{cf}\right)$ at the fusion center and the new information received by the tracker, the data fusion information of the fusion center at time $t_{k}$ can be calculated:(45)$$\Upsilon \left(t_{k}\right)=\Upsilon \left(t_{k},t_{cf}\right)+\Delta \Upsilon^{1}\left(t_{k}\right)+\Delta \Upsilon^{2}\left(t_{k}\right),$$
(46)$$\hat{\gamma }\left(t_{k}\right)=\hat{\gamma }\left(t_{k},t_{cf}\right)+\Delta {\hat{\gamma }}^{1}\left(t_{k}\right)+\Delta {\hat{\gamma }}^{2}\left(t_{k}\right),$$

Then, the fusion covariance $P^{IF}\left(t_{k}\right)$ and state estimate ${\hat{x}}^{IF}\left(t_{k}\right)$ can be computed.
(47)$$P^{IF}\left(t_{k}\right)=\Upsilon^{-1}\left(t_{k}\right),{\hat{x}}^{IF}\left(t_{k}\right)=P^{IF}\left(t_{k}\right)\hat{\gamma }\left(t_{k}\right),$$
and if, at this time, $t_{k}=t_{f}$, the fusion ends; otherwise, let $t_{cf}=t_{k}$ and start to predict the relevant information from Formula (39).

## 4. Simulation Verification and Analysis

In order to verify the effectiveness of the aforementioned autonomous-vehicle multitarget-tracking method based on the fusion of millimeter-wave radar and lidar information, it was necessary to conduct simulation experiments, and the performance of the multitarget-tracking method proposed in this paper was verified by using the Generalized Optimal SubPattern Assignment Metric (GOSPA) [43]. The high-speed driving scene of the autonomous vehicle and the target vehicle is shown in Figure 3, in which there is one autonomous vehicle and four target vehicles.

The autonomous vehicle is located in the middle lane of the three lanes, and there are two target vehicles in front of the autonomous vehicle, which are located in the middle lane and the right lane, respectively; behind the ego vehicle, there is a target car in the center lane and another target car in the left lane. The autonomous vehicle is equipped with four millimeter-wave radars and one lidar, and the radar detection coverage overlaps. At the beginning of the simulation, the autonomous vehicle and target vehicles 1, 3, and 4 travel at a speed of 25 m/s, and target vehicle 2 travels at a speed of 35 m/s. It can be seen that the simulation scene includes a car-following scene and an overtaking scene.

The parameters of the autonomous vehicle, target vehicle, and sensors are shown in Table 1 and Table 2.

We carried out a performance analysis of a single-radar tracker (such as the millimeter-wave radar tracker and lidar tracker) and fusion estimation tracker from two aspects of the simulation visualization interface and quantitative indicators, and judged the comprehensive performance of the fusion estimation tracker based on different events in the scene.

### 4.1. Simulation Scene Visualization Analysis

Figure 4 is the visual interface of the simulation scene of the millimeter-wave radar, lidar, and track fusion; the interface consists of five figures, namely the symbol feature figure, the front view scene figure, the rear-view scene figure, the single-sensor figure, and the close-up figure. The symbolic feature graph explains the target information of different types of sensors. The front view scene graph and the rear-view scene graph are the forward scene and the backward scene at the same moment under the condition of the third perspective, respectively. The single-sensor figure shows only target-tracking information from millimeter-wave radar or lidar, while the close-up figure shows key information about the target vehicle. The prefixes “R”, “L”, and “F” represent the target information of the millimeter-wave radar tracker, lidar tracker, and fusion estimation tracker, respectively, and the Arabic numerals represent the unique identifier.

The millimeter-wave radar can obtain multiple pieces of detection information of the same target vehicle, which seriously affects the target detection effect. After being processed by the extended target tracker, the detection effect is greatly improved. The scene visualization interface with a simulation time step of 50 is shown in Figure 4a. It can be seen from the figure that whether it is millimeter-wave radar, lidar, or fusion estimation tracker, they can all effectively track four target vehicles and generate the respective target tracks.

It can be seen from Section 3.1 that when using lidar for target tracking, the Euclidean algorithm will be used to cluster the target point cloud in this state, and then the state vector of the bounding box detector will be obtained according to the clustering results. However, when the distance between different detection objects is very close, the effect of the clustering algorithm will be affected. The scene visualization interface with a simulation time step of 70 is shown in Figure 4b. It can be seen from the figure that the distance between target vehicle 2 and target vehicle 3 in front of the autonomous vehicle is very close; at this point, the bounding box detector aggregates the point cloud of each vehicle into a larger bounding box, and as a result, the target point detected by the lidar deviates from the center of the vehicle. Compared with the lidar tracker, the target points detected by the fusion estimation tracker are highly consistent with the center of the vehicle.

### 4.2. Quantitative Index Analysis

We used quantitative indicators to evaluate the tracking performance of single-radar trackers (such as millimeter-wave radar trackers and lidar trackers) and fusion estimation trackers, and analyzed the performance of different trackers in simulation scenarios. First, the tracking performance of the millimeter-wave radar, lidar, and fused estimation trackers was evaluated using estimation errors for position, velocity, size, and orientation.

The average estimation errors of millimeter-wave radar, LiDAR, and fusion estimation tracker are shown in Figure 5. It can be seen from the figure that the tracking performance of the fusion estimation tracker for the target vehicle was significantly better than that of the single-radar trackers in terms of the tracking effects of different targets; the estimation errors of position, velocity, size, and direction were reduced by 85.5%, 64.6%, 75.3%, and 9.5%, respectively. It should be noted that since the millimeter-wave radar had no size and direction information, it is not shown in the figure.

The tracking performance or localization accuracy of different trackers can be quantitatively evaluated with the GOSPA index at each time step, with lower index values indicating higher tracking accuracy. The tracking performance of millimeter-wave radar, lidar, and track fusion trackers was quantitatively measured using the “missing target” and “false tracking” components of the GOSPA metric. As shown in Figure 6, the “lost target” dropped to 0 after the tracking system ran for 8 time steps. The reason for the high value at the beginning of the period was due to the delay of the system. The “fault trace” component was always 0, thus indicating that the trace system did not generate fault traces. When the simulation time step was 70, the GOSPA index value of the lidar increased significantly; the main reason is that the distance between different detection targets was too close. The average value of the GOSPA index obtained using the fusion estimation method was 19.8% lower than that of the single-type sensor, which shows that the overall tracking accuracy was effectively improved after the information fusion of different sensors. The average value of the GOSPA index obtained using the fusion estimation method was 19.8% lower than that of the single-type sensor, which shows that the overall tracking accuracy was effectively improved after the information fusion of different sensors.

Through the performance analysis of the single-radar tracker (such as the millimeter-wave radar tracker and the lidar tracker) and the fusion estimation tracker through the two aspects of the simulation visual interface and the quantitative index, it can be concluded that the multitarget-tracking method for autonomous vehicles based on millimeter-wave radar and lidar proposed in this paper can realize the track tracking of multiple target vehicles in high-speed driving scenarios. Compared with the single-radar tracker, the fusion estimation tracker can effectively avoid the detection target point deviation problem caused by the too-close distance between different detection targets.

## 5. Conclusions

Most of the current autonomous-vehicle target-tracking methods based on the fusion of millimeter-wave radar and lidar information struggle to guarantee accuracy and reliability in the measured data, and cannot effectively solve the multitarget-tracking problem in complex scenes. This paper takes the distributed multisensor multitarget tracking (DMMT) system as the basic framework, based on millimeter-wave radar and lidar; a multitarget-tracking method for autonomous vehicles is proposed that comprehensively considers key technologies such as target tracking, sensor registration, track correlation, and data fusion. In order to verify the effectiveness of the proposed method, a performance analysis of the single-radar tracker (such as the millimeter-wave radar tracker and the lidar tracker) and the fusion estimation tracker is carried out from two aspects of the simulation visualization interface and quantitative indicators. The multitarget-tracking method for autonomous vehicles based on millimeter-wave radar and lidar proposed in this paper can realize the track tracking of multiple target vehicles in high-speed driving scenarios. Compared with the single-radar tracker, the position, velocity, size, and direction estimation errors of the track fusion tracker are reduced by 85.5%, 64.6%, 75.3%, and 9.5%, respectively, and the average value of GOSPA indicators is reduced by 19.8%. The fusion estimation tracker can effectively avoid the detection target point deviation problem caused by the too-close distance between different detection targets, and can obtain more accurate target state information than the single-radar tracker. In future research, we will further consider scenarios such as rainy and snowy days, viaducts, tunnels, etc. and use different combinations of millimeter-wave radar, lidar, and cameras to verify the effectiveness of the proposed method in the track tracking of multiple targets by way of combining a simulation and a real vehicle experiment.

## Figures and Tables

**Figure 1 sensors-23-06920-f001:**
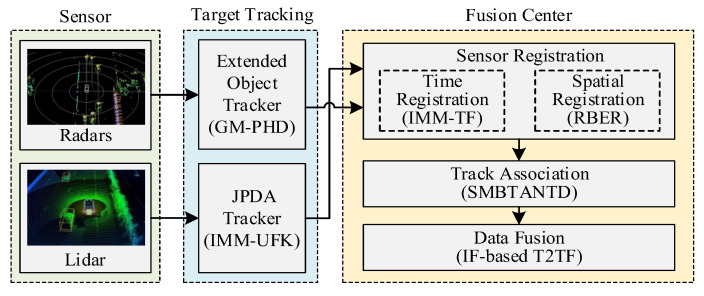
Flow chart of multisensor multitarget tracking.

**Figure 2 sensors-23-06920-f002:**
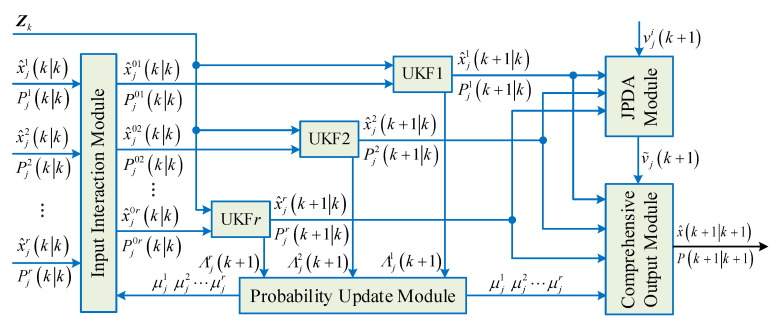
JPDA tracker configured with IMM-UKF.

**Figure 3 sensors-23-06920-f003:**
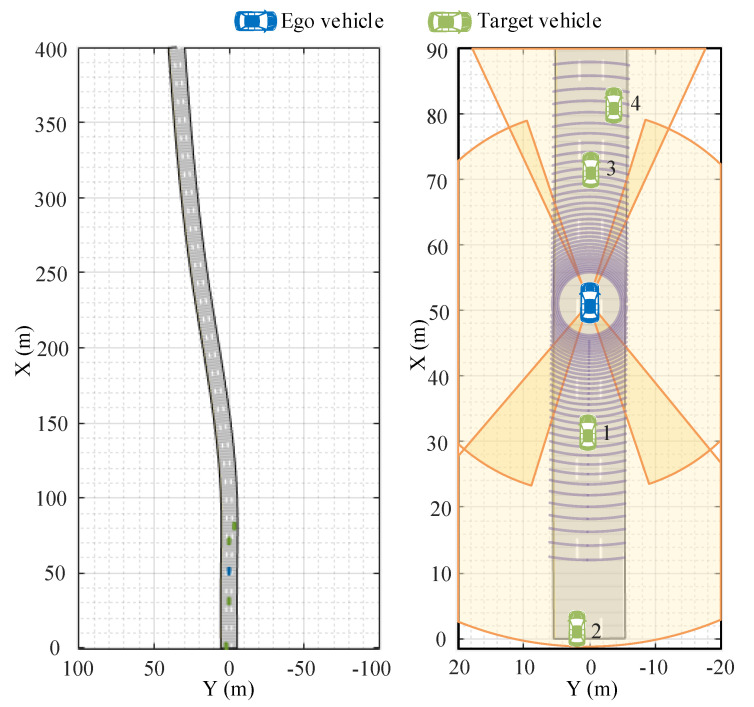
High−speed driving scene of autonomous vehicle and target vehicle.

**Figure 4 sensors-23-06920-f004:**
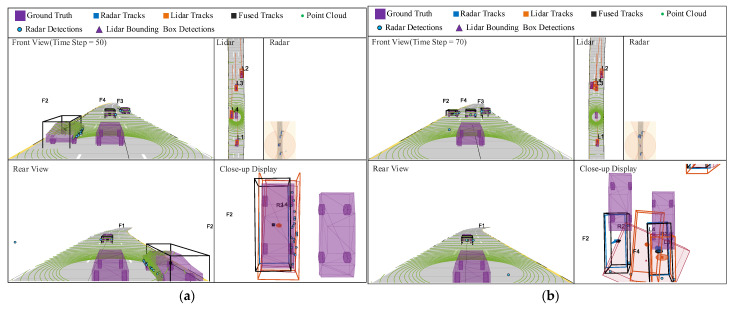
Radar, lidar, and the track-level fusion simulation scenarios. (**a**) Track maintenance; (**b**) closely spaced targets.

**Figure 5 sensors-23-06920-f005:**
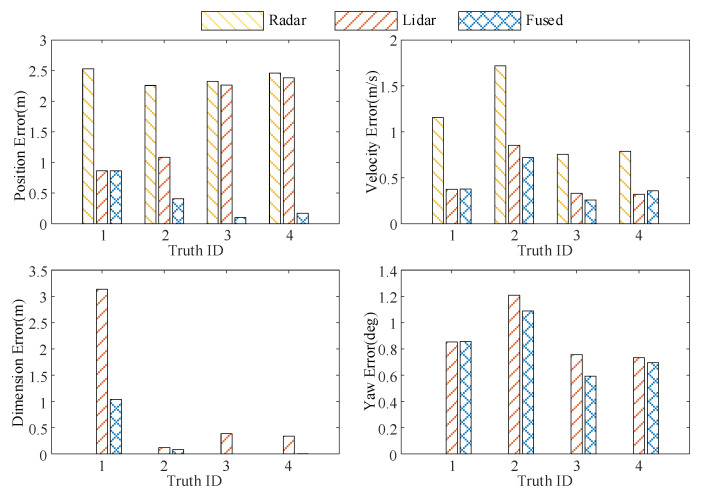
Average estimation error of different trackers.

**Figure 6 sensors-23-06920-f006:**
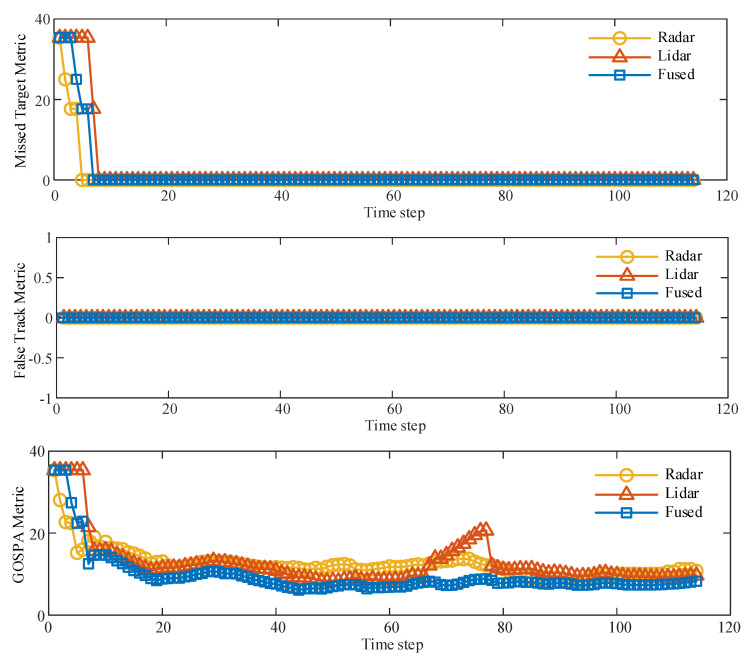
GOSPA tracking performance evaluation.

**Table 1 sensors-23-06920-t001:** Motion parameter setting table.

Parameter	Ego Vehicle	Target Vehicle 1	Target Vehicle 2	Target Vehicle 3	Target Vehicle 4
Speed (m/s)	25	25	35	25	25
Length (m)	4.7	4.7	4.7	4.7	4.7
Width (m)	1.8	1.8	1.8	1.8	1.8
Height (m)	1.4	1.4	1.4	1.4	1.4
Road centers	[0, 0; 50, 0; 100, 0; 250, 20; 400, 35]				
Sample time (s)	0.1				
Lane specifications	3.0				

**Table 2 sensors-23-06920-t002:** Sensor setting table.

	Front Radar	Rear Radar	Left Radar	Right Radar	LiDAR
Mounting location (m)	(3.7, 0.0, 0.6)	(−1.0, 0.0, 0.6)	(1.3, 0.9, 0.6)	(1.3, −0.9, 0.6)	(3.7, 0.0, 0.6)
Mounting angles (deg)	(0.0, 0.0, 0.0)	(180.0, 0.0, 0.0)	(90.0, 0.0, 0.0)	(−90.0, 0.0, 0.0)	(0.0, 0.0, 2.0)
Horizontal field of view (deg)	(5.0, 30.0)	(5.0, 90.0)	(5.0, 160.0)	(5.0, 160.0)	(5.0, 360.0)
Vertical field of view (deg)	-	-	-	-	(5.0 40.0)
Azimuth resolution (deg)	6.0	6.0	6.0	6.0	0.2
Range limits (m)	(0.0, 250.0)	(0.0, 100.0)	(0.0, 30.0)	(0.0, 30.0)	(0.0, 200.0)
Range resolution (m)	2.5	2.5	2.5	2.5	1.25

## Data Availability

Data supporting this systematic review are available in the References Section. In addition, the data presented in this study are available upon request from the corresponding author.

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
