# Peer review of "Multitarget-Tracking Method Based on the Fusion of Millimeter-Wave Radar and LiDAR Sensor Information for Autonomous Vehicles"

_sensors, 2023, doi:10.3390/s23156920_

Round 1

Reviewer 1 Report

This paper proposed an MOT approach for autonomous driving applications based on fusion of two commonly used sensors on autonomous vehicles -- millimeter wave radar and LiDAR. This topic is very interesting and can be of high significance and applicability for future production vehicles. Generally, this paper is well-written and easy to follow. But the following concerns should be carefully addressed, before it can be considered for publication. 

1. The major contributions of this work are not clearly stated. It is recommended that this issue be properly addressed in the revised manuscript.

2. This work is based on two types of sensors, i.e. millimeter wave radar and LiDAR, which is reasonable as these two sensors are frequently used. Apart from these two, camera is also a very commonly used type of on-board sensors. It is suggested that the authors explain the reasons why camera is not involved in this work.

3. In practical applications, orbit correlation and spatial registration are usually coupled. Incorrect trajectory correlation can lead to spatial registration misalignment, which can interfere with data collection and lead to chaotic trajectory correlation. I did not see a solution to the coupling problem between orbit correlation and spatial registration in this article. Please explain.

4. In the conclusion section of the article, a detailed and objective summary is provided for the entire paper. Do the authors have any further research plans and limitations of the methods presented in this article? If there are any, it is recommended that the authors supplement them in the next version.

In general, the English language of this paper is satisfactory. It is recommended that the authors perform a thorough proofreading in order to fix any typos or grammar mistakes in the manuscript. Also, long sentences are suggested to be truncated to enhance readability of the paper.

Reviewer 2 Report

This manuscript presents a multi-target tracking method for autonomous vehicles that comprehen-sively considers key technologies such as target tracking, sensor registration, track association and data fusion based on millimeter-wave radar and lidar. Please consider the following comments for its improvement.

1. There are some references which are not in your list to look at (Ji-Eun Joo,Shinhae Choi,Yeojin Chon,A Low-Cost Measurement Methodology for LiDAR Receiver Integrated Circuits”ï¼›Jubong Lee,,Jinseo Hong,Kyihwan Park,Frequency Modulation Control of an FMCW LiDAR Using a Frequency-to-Voltage Converter , these two as an example).

2. The paper is in general sufficiently well written and structured, but still requires a considerable spell and grammar check. There are many small typos that need to be corrected.

3. In "3.1. Single Sensor Target Tracking", extended target trackers are used for millimeter wave radar, while traditional JPDA trackers are used for lidar. Why do we use different track trackers for different sensors?

4. Space registration and track association are important components in multi-sensor information fusion, and the quality of space registration and track association directly affects the performance of subsequent fusion. In most spatial registration algorithms, it is assumed that the problem of track association has been solved. Similarly, in the track association algorithm, spatial registration is completed by default. In practical applications, track correlation and spatial registration are often coupled. Incorrect track association leads to inaccurate spatial registration, which can interfere with data acquisition and lead to chaotic track association. How is the coupling problem between track correlation and spatial registration resolved?

Please carefully check the spelling errors in the manuscript

Reviewer 3 Report

In the article under review, the authors present the results of a study aimed at improving the efficiency of tracking the road space around autonomous vehicles. The authors propose a method for tracking multiple target vehicles in high-speed driving scenarios based on an algorithm for combining tracking information provided by millimeter wave radar and lidar.

In the Introduction and literature review, the prerequisites for conducting research are considered in sufficient detail, and the purpose of the paper is formulated. In the main parts of the paper, a description of the program framework of multi-sensor multi-target tracking is presented. A detailed mathematical description of target tracking using separate sensors and a mathematical description of the heterogeneous sensor data fusion are given. The results of simulation experiments and their analysis are presented.

I would like to thank the authors for the quality research and congratulate the authors on a well-prepared paper, however, I draw the attention of the authors to the following shortcomings, the correction of which would improve the quality of the paper, and I would also like to ask for clarifications:

  1. There is an inaccuracy in Table 1: the speed of target vehicle 2 must be 35 m/s (as indicated in line 491).
  2. In the paper, the authors presented only the results of the simulation and did not provide information on the verification of the results obtained. What is the basis of the authors' confidence in the adequacy of the obtained results?
  3. In the Conclusion section, I suggest that the author not only sum up the results but also indicate what, from their point of view, should be the ways for the further development of their research.

In general, I recommend this article for publication after minor revision.

Round 2

Reviewer 2 Report

The author responded to all questions and revised the manuscript. This version of the manuscript is suggested to be considered for acceptance.